# Risk Assessment of Postoperative Pneumonia in Cancer Patients Using a Common Data Model

**DOI:** 10.3390/cancers14235988

**Published:** 2022-12-04

**Authors:** Yong Hoon Lee, Do-Hoon Kim, Jisun Kim, Jaetae Lee

**Affiliations:** 1Department of Internal Medicine, School of Medicine, Kyungpook National University, Daegu 41944, Republic of Korea; 2Medical Big Data Research Center, Kyungpook National University Hospital, Daegu 41944, Republic of Korea; 3Department of Medical Informatics, Kyungpook National University Hospital, Daegu 41944, Republic of Korea; 4Department of Nuclear Medicine, School of Medicine, Kyungpook National University, Daegu 41944, Republic of Korea

**Keywords:** postoperative pneumonia, cancer, common data model

## Abstract

**Simple Summary:**

The incidence of postoperative pneumonia (POP) in patients with cancer is high owing to functional impairment of the immune system associated with cancer and additional lung damage caused by surgery. The incidence of POP following major cancer surgeries is unclear. Therefore, we investigated the incidence and risk factors of POP after cancer surgery in patients with the five most common cancers in Korea using a common data model (CDM). The CDM was a method to easily solve clinical questions using medical big data. The cumulative POP incidence in the five cancers using the CDM was approximately 3%. POP was most common in lung cancer (*n* = 91, 4.5%), followed by gastric (*n* = 133, 3.3%), colon (*n* = 19, 3.1%), liver (*n* = 14, 1.7%), and breast (*n* = 5, 0.5%) cancers. Older age, male sex, chronic pulmonary disease, mood disorder, and cerebrovascular disease were POP risk factors in patients with cancer.

**Abstract:**

The incidence of postoperative pneumonia (POP) in patients with cancer is high, but its incidence following major cancer surgeries is unclear. Therefore, we investigated the incidence and risk factors of POP after surgery in patients with the five most common cancers in Korea using a common data model (CDM). Patients aged >19 years who underwent gastric, colon, liver, lung, or breast cancer surgery between January 2011 and December 2020 were included, excluding patients who underwent chemotherapy or radiotherapy. Pneumonia was defined as a pneumonia diagnosis code in patients hospitalized postoperatively. Gastric, colon, lung, breast, and liver cancers were noted in 4004 (47.4%), 622 (7.4%), 2022 (24%), 958 (11.3%), and 839 (9.9%) of 8445 patients, respectively. The cumulative POP incidence was 3.1% (*n* = 262), with the highest incidence in lung cancer (*n* = 91, 4.5%), followed by gastric (*n* = 133, 3.3%), colon (*n* = 19, 3.1%), liver (*n* = 14, 1.7%), and breast (*n* = 5, 0.5%) cancers. In multivariable analysis, older age, male sex, history of chronic pulmonary disease, mood disorder, and cerebrovascular disease were POP predictors. The cumulative POP incidence in the five cancers using the CDM was approximately 3%. Older age, male sex, chronic pulmonary disease, mood disorder, and cerebrovascular disease were POP risk factors in patients with cancer.

## 1. Introduction

Postoperative pneumonia (POP) is associated with increased morbidity, mortality, and costs [1,2]. This potentially life-threatening complication occurs in up to 9% of patients undergoing high-risk surgery [3]. Postoperative pulmonary complications, including lung injury, can also result in the increased consumption of medical resources and longer stays in the intensive care unit and hospital [4]. Despite advances in surgical and anesthetic techniques, POP remains a common and important complication following surgery [5].

Various factors related to cancer or its treatment contribute to immune dysfunction, thereby promoting the development of pneumonia in cancer patients [6]. Additionally, as a treatment modality, surgery can lead to lung injury [4]. Cancer affects not only the occurrence of pneumonia but also the clinical outcomes. Pneumonia could contribute to cancer death, directly, by causing gas exchange impairment, systemic inflammation, and sepsis, or indirectly, by interfering with antineoplastic therapies [7]. The incidence of POP following cancer surgery has been reported to differ according to the type of cancer, and a recent study on lung cancer showed that the incidence of POP reached approximately 15% [8]. Another study on patients undergoing major cancer surgery showed that the incidence of POP gradually increased over time and that pneumonia increased the mortality risk by 6.3 times [9]. Thus, it is important to assess the risk of pneumonia after cancer surgery in advance; however, data based on the comprehensive analysis of various cancers for POP are still lacking.

The common data model (CDM) has been used to standardize medical records and data for multicenter collaborative studies. The CDM provides medical information related to patient care, such as diagnosis, medication, surgery, and examination. Thus, it has been used to create big data of clinical information suitable for conducting studies. In addition, since the patient’s personal information is anonymized, the CDM does not violate the guidelines for research ethics. Because the CDM is related to large-scale medical data, it can be used to evaluate rare complications of rare diseases. Many advantages of CDM-related research have been noted, and the CDM is gaining popularity in various medical fields [10,11,12,13,14,15].

Therefore, our goal was to investigate the incidence and risk factors of POP after cancer surgery in patients with the five most common cancers (gastric, colon, lung, breast, and liver cancers) in Korea using the CDM.

## 2. Materials and Methods

This study was approved by the institutional review board of Kyungpook National University Hospital (KNUH IRB No. 2022-04-010) and performed in accordance with the Declaration of Helsinki (1989) by the World Medical Association.

### 2.1. Data Source

This retrospective observational study was conducted at Kyungpook National University Hospital. We used the Observational Health Data Sciences and Informatics (OHDSI) open-source software and the Observational Medical Outcomes Partnership (OMOP) CDM version 5.3 database, into which the electronic medical records (EMR) data were transformed. The database contained deidentified EMR data converted to a standard format using the OMOP CDM system. Over a 15-year duration beginning in 2006, the EMR data of more than two million patients were transformed and collected from our hospital’s CDM database.

### 2.2. Registry-Based Patient Selection

In this retrospective registry-based case series, we investigated a cohort of patients who underwent cancer surgery between January 2011 and December 2020 using the CDM database. We identified 17,928 cancer patients (gastric: 5959, colon: 979, lung: 3866, breast: 5201, and liver: 1923) in the registry. The cohort registry collects information on patient demographics. Among the patients included in the enrollment, patients who had a previous pneumonia diagnosis (*n* = 100) and patients who had undergone chemotherapy or radiotherapy (*n* = 9383) were excluded. Finally, 8445 patients were included in the analysis (Figure 1).

### 2.3. Variables

We retrospectively collected the following clinical data from the CDM database: demographic variables (age and sex), comorbidities, type of cancer, date of surgery, and development of POP. Pneumonia, as an outcome, was defined as a pneumonia diagnosis code in a patient who was hospitalized after surgery.

### 2.4. Statistical Analyses

Statistical analysis was performed using the R software (version 4.0.4; R Foundation for Statistical Computing, Vienna, Austria). Descriptive data are expressed as mean ± standard deviation (SD), median (min-max), or number and frequency, where applicable. The chi-square and Fisher exact tests were used to analyze categorical variables. T-tests were used to analyze continuous variables. Univariate and multivariate Cox proportional hazard regression analyses were used to identify potential risk factors for POP incidence in patients with cancer. Statistical significance was set at *p* < 0.05.

## 3. Results

### 3.1. Baseline and Demographic Data

Among the 8445 patients, gastric, colon, lung, breast, and liver cancer were noted in 4004 (47.4%), 622 (7.4%), 2022 (24%), 958 (11.3%), and 839 (9.9%) patients, respectively. There were statistically significant differences in age, sex, hypertension, chronic pulmonary disease, mood disorders, Parkinson’s disease, and cerebral vascular disease between the POP and non-POP groups (Table 1).

### 3.2. Pneumonia Incidence

The cumulative POP incidence was 3.1% (262 patients). POP was most common in lung cancer (91 patients, 4.5%), followed by gastric (133 patients, 3.3%), colon (19 patients, 3.1%), liver (14 patients, 1.7%), and breast (five patients, 0.5%) cancer (Table 2). As shown in Table 2, the hazard ratios (95% confidence interval) for the development of POP in patients with lung, gastric, colon, and liver cancer were 8.08 (3.29–19.89), 5.67 (2.32–13.85), 6.14 (2.29–16.44), and 3.22 (1.16–8.94), respectively, with breast cancer as the reference.

The occurrence of POP with respect to time for gastric, colon, lung, liver, and breast cancer is shown graphically (Figure 2). Significant differences were found between the five groups (*p* < 0.001). Kaplan–Meier analysis showed that patients with lung cancer developed POP more frequently than those with other cancers (log-rank *p* < 0.001).

### 3.3. Risk Factors for POP

In the univariate Cox regression analysis, the following variables were associated with POP: older age, male sex, hypertension, chronic pulmonary disease, mood disorder, Parkinson’s disease, and cerebrovascular disease (Table 3). In the multivariate analysis, older age, male sex, chronic pulmonary disease, mood disorder, and cerebrovascular disease were related to POP development (Table 3).

Appendix A shows the results of the subgroup analysis of POP risk factors for lung and gastric cancers. In univariate Cox regression analysis, POP-related variables associated with lung cancer were old age, male sex, hypertension, chronic pulmonary disease, mood disorder, and cerebrovascular disease. However, in multivariate analysis, only old age, mood disorder, and cerebrovascular disease were associated with POP occurrence. In univariate Cox regression analysis of gastric cancer, the variables associated with POP were old age, male sex, hypertension, renal disease, chronic pulmonary disease, mood disorder, Parkinson’s disease, and cerebrovascular disease. In addition, a multivariate analysis of gastric cancer showed that older age, male sex, chronic pulmonary disease, and mood disorder were associated with POP occurrence.

## 4. Discussion

The present study aimed to investigate the incidence and risk factors of POP in patients undergoing major cancer surgery. Overall, the cumulative incidence of POP was approximately 3%, and POP incidence was the highest among patients with lung cancer undergoing surgery. Older age, male sex, chronic pulmonary disease, mood disorder, and cerebrovascular disease were independently associated with POP development.

Information is scarce regarding the integrated analysis of POP in various types of cancer, and the reported incidence of POP is somewhat similar to our results. In a previous study by Trinh et al., which analyzed the data of approximately 2.5 million cancer patients undergoing eight types of surgery, the overall rate of in-hospital POP was 3.5% [9]. Another study by Jung et al., which included patients with the five most common cancers, as in our study, reported that the 1-year cumulative incidence of POP was 2% [16]. They also showed that lung cancer surgery was the most common precedent for POP [16], which is consistent with our results. Although it is difficult to elucidate a clear mechanism from our data alone, lung cancer patients seem to be more vulnerable to POP than other cancer patients, especially in terms of anatomical distortion; architectural changes in the lung due to surgical intervention or radiotherapy could predispose patients to infection with bacterial pathogens [6]. Incompletely resected tumors or recurrent tumors can cause airway obstruction, which may impair mucociliary clearance and lead to pneumonia [6]. In addition, since chronic pulmonary disease is a risk factor for lung cancer and shares risk factors such as smoking [17], the likelihood that patients with lung cancer have pre-existing chronic lung disease is higher than that of other cancers. Chronic pulmonary disease can also increase the risk of bacterial colonization of the airways and pneumonia [18,19].

Chronic pulmonary disease was an independent risk factor for POP in the current study. Several studies have suggested that chronic pulmonary disease, including chronic obstructive pulmonary disease, is a predictor of POP in cancer patients, and most of these studies enrolled patients with lung cancer [19,20,21]. A reduced forced expiratory volume in 1 s/forced vital capacity (FEV1/FVC) ratio, an indicator of airway obstruction, has also been reported to be significantly associated with pneumonia after lung cancer surgery [22]. Chronic obstructive pulmonary disease and its treatment are also well-known risk factors for pneumonia in the general population [23]. Considering our results, chronic pulmonary disease may have had a significant effect on POP in patients with lung cancer and other cancers.

Several studies have shown a significant relationship between male sex and POP in cancer patients. In previous studies of patients with oral and gastric cancer, the male sex was found to be an independent risk factor for POP [24,25]. In addition, the study by Trinh et al., which included several cancer types, showed that male patients had a higher risk of POP than female patients [9], which is in line with our results. Pneumonia, whether community-acquired or nosocomial, is generally more common in males, and several explanations for this occurrence have been proposed [26]. Biologically, the action of sex hormones is different; estrogen can act as an immune stimulant, while androgen has an anti-inflammatory effect [27]. Additionally, lifestyle, behavioral, and socioeconomic differences have been highlighted as factors that could explain the sex difference in pneumonia incidence [26].

Interestingly, mood disorders were also associated with increased odds of POP in our study. To our knowledge, no published study has identified mood disorder as a risk factor for pneumonia after cancer surgery. In general, patients with mental illness are at an increased risk of various postoperative complications, including pneumonia [28]. Patients with severe mental illness are more likely to have poor insight into their overall health and difficulty in self-care, which can negatively affect hygiene [29]. The prevalence of smoking, alcohol, and substance abuse, which are known risk factors for pulmonary infection, is higher in patients with severe mental illness than in the general population and may contribute to the risk of pneumonia [30]. In addition, the hematologic side effects of drugs used for treating mental illness can also promote infection [31]. Although mood-related disorders are not uncommon in patients with cancer, they are often overlooked in clinical settings [32]. Our results suggest that screening for mood disorders in patients with cancer undergoing surgery is important for the early diagnosis or prevention of POP.

The CDM uses the OHDSI database, which aims to facilitate reproducible global, large-scale observational research. CDMs have been developed to enable the management of large amounts of data in the medical field. To assess the effectiveness of our methodology, we used CDM coding algorithms. Tertiary centers that use the CDM use the same electronic medical records; therefore, we queried their CDM databases to extract the data of interest. CDM-related research in Korea is still in its infancy, and multi-institutional databases have not been translated into common and simple terms. Therefore, it is difficult to conduct multicenter research using the advantages of the CDM in Korea. However, if a database is actively built, this should be possible.

Our study had several limitations. First, selection bias could not be avoided because of the retrospective nature of the study. Second, diagnostic code entries may have been omitted in some POP cases; thus, the incidence of POP in our data may have been underestimated. Third, data on common risk factors for pneumonia, such as smoking, alcoholism, and immunocompromised status, were not available. In addition, we could not obtain detailed information on the type of surgery, the volume of surgery, and the stage of the tumor on the incidence of POP. These factors might have acted as confounders.

## 5. Conclusions

The cumulative incidence of POP in five cancers calculated using a CDM was approximately 3%. Older age, male sex, chronic pulmonary disease, mood disorder, and cerebrovascular disease were associated with an increased risk of POP in patients with cancer. Additional multicenter studies using a CDM are needed to validate our findings.

## Figures and Tables

**Figure 1 cancers-14-05988-f001:**
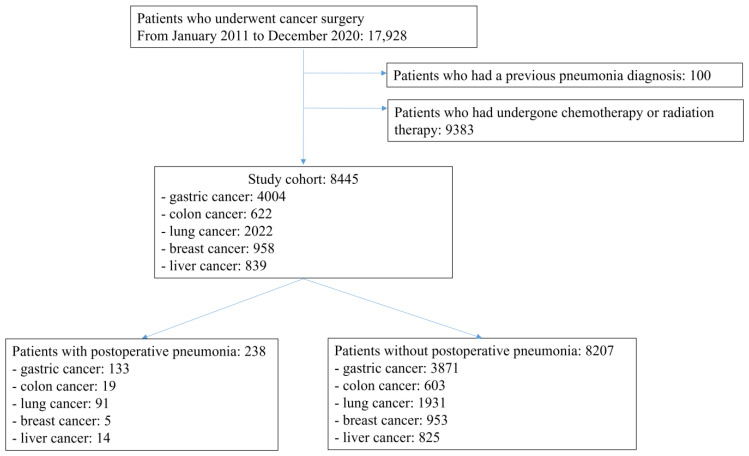
Flow diagram of the study.

**Figure 2 cancers-14-05988-f002:**
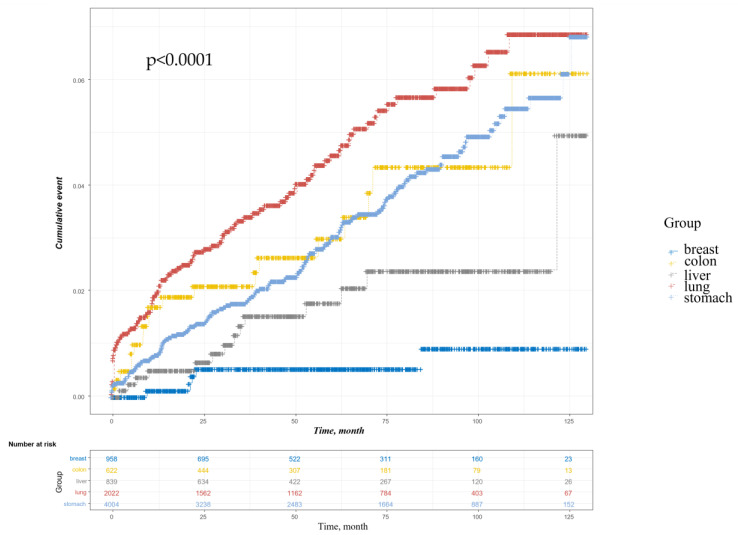
Cumulative event curves for postoperative pneumonia according to cancer type.

**Table 1 cancers-14-05988-t001:** Patient demographic data.

Variables	Non-POP Group (*n* = 8183)	POP Group (*n* = 262)	*p*-Value
Sex *			<0.001
Female	3596 (43.9)	64 (24.4)	
Male	4587 (56.1)	198 (75.6)	
Age (years) *	60.67 ± 14.10	68.69 ± 10.22	<0.001
Hypertension *	320 (3.9)	23 (8.8)	<0.001
Diabetes mellitus	481 (5.9)	22 (8.4)	0.118
Renal disease	138 (1.7)	8 (3.1)	0.153
Mild liver disease	390 (4.8)	10 (3.8)	0.573
Heart failure	59 (0.7)	4 (1.5)	0.26
Chronic pulmonary disease *	329 (4.0)	29 (11.1)	<0.001
Mood disorder *	63 (0.8)	7 (2.7)	0.003
Dementia	16 (0.2)	2 (0.8)	0.2
Hypothyroidism	96 (1.2)	2 (0.8)	0.751
Parkinson’s disease *	26 (0.3)	4 (1.5)	0.007
Peripheral vascular disease	64 (0.8)	3 (1.1)	0.766
Peptic ulcer disease	271 (3.3)	12 (4.6)	0.343
Cerebrovascular disease *	357 (4.4)	28 (10.7)	<0.001

Values are presented as *n* (%) or mean ± standard deviation. * *p* < 0.05. POP: postoperative pneumonia.

**Table 2 cancers-14-05988-t002:** Cumulative incidence of postoperative pneumonia in all patients according to cancer type.

	No. of Patients	No. of POP Cases	HR (95% CI)	*p*-Value
Total	8445	238		
Gastric cancer	4004	133	5.67 (2.32–13.85)	<0.001
Colon cancer	622	19	6.14 (2.29–16.44)	<0.001
Lung cancer	2022	91	8.08 (3.29–19.89)	<0.001
Liver cancer	839	14	3.22 (1.16–8.94)	0.025
Breast cancer	958	5	Reference	

Based on univariable Cox proportional hazard regression analysis. POP: postoperative pneumonia; HR: hazard ratio; CI: confidence interval.

**Table 3 cancers-14-05988-t003:** Risk factors for postoperative pneumonia.

Variables	Univariate		Multivariate	
	Hazard Ratio (95% CI)	*p*-Value	Hazard Ratio (95% CI)	*p*-Value
Age	1.06 (1.05–1.07)	<0.001	1.05 (1.04–1.06)	<0.001
Sex (Ref: Female)	2.32 (1.75–3.07)	<0.001	2.19 (1.65–2.91)	<0.001
Hypertension	2.04 (1.33–3.13)	0.001		
Chronic pulmonary disease	3.11 (2.11–4.58)	<0.001	2.10 (1.42–3.10)	<0.001
Mood disorder	3.84 (1.81–8.14)	<0.001	3.16 (1.46–6.83)	0.003
Parkinson’s disease	4.46 (1.66–11.98)	0.003		
Cerebrovascular disease	3.18 (2.15–4.72)	<0.001	1.84 (1.22–2.77)	0.004

CI: confidence interval.

## Data Availability

The obtained access to the common data model database from the Department of Medical Information at Kyungpook National University Hospital within the research project (IRB number: 2022-04-010). For legal reasons, the full dataset cannot be published. However, access to the common data model database for research purposes can be requested from the Department of Medical Information at Kyungpook National University Hospital.

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
