# Peer review of "Risk Assessment of Postoperative Pneumonia in Cancer Patients Using a Common Data Model"

_cancers, 2022, doi:10.3390/cancers14235988_

Round 1

Reviewer 1 Report

I have no additional questiоns directly for authors. The authors analyze the frequency and risk factors of postoperative
pneumonia in cancer patients. A large clinical material was used - about 18 thousand patients, of which about 8500 cases were included in the final analysis
. Most of the patients suffered from cancer of the stomach, lung and breast. Postoperative pneumonia (POP) was observed in 3.1%. Unfortunately, the authors do not analyze the reasons of the development of postoperative pneumonia, there is no data on the effect of the volume of surgery and the stage of the tumor on the incidence of POP. The risk factors for POP identified by the authors (old age, male gender, comorbid pathology, especially, primarily chronic lung diseases, operations for lung cancer) are well known, this information is not of great scientific interest. The authors used 24 literary sources, of which 50% - 13 articles were published before 2016.

Author Response

I have no additional questiоns directly for authors. The authors analyze the frequency and risk factors of postoperative pneumonia in cancer patients. A large clinical material was used - about 18 thousand patients, of which about 8500 cases were included in the final analysis. Most of the patients suffered from cancer of the stomach, lung and breast. Postoperative pneumonia (POP) was observed in 3.1%. Unfortunately, the authors do not analyze the reasons of the development of postoperative pneumonia, there is no data on the effect of the volume of surgery and the stage of the tumor on the incidence of POP. The risk factors for POP identified by the authors (old age, male gender, comorbid pathology, especially, primarily chronic lung diseases, operations for lung cancer) are well known, this information is not of great scientific interest. The authors used 24 literary sources, of which 50% - 13 articles were published before 2016.

Reply: We thank the reviewer for reviewing our manuscript. We agree with your comment. In our study, we were unable to analyze the cause of postoperative pneumonia due to the limitations of available data from common data model. In the currently established common data model, it was not possible to obtain information on the amount of surgery and the stage of the tumor. We have considered conducting an in-depth analysis of the cause of postoperative pneumonia and analysis using a database that provides more detailed information than common data model. Based on your advice, we have modified the limitation of the discussion as follows:

“we could not obtain detailed information on the type of surgery, the volume of surgery, and the stage of the tumor on the incidence of POP. These factors might have acted as confounders.”

Regarding references, we have cited six new articles on CDM research published in the last 5 years in the introduction section.

List of newly added references

  1. Kim, D.H.; Lee, J.E.; Kim, Y.G.; Lee, Y.; Seo, D.W.; Lee, K.H.; Lee, J.H.; Kim, W.S.; Kim, Y.H.; Oh, J.S. High-Throughput Algorithm for Discovering New Drug Indications by Utilizing Large-Scale Electronic Medical Record Data. Clin Pharmacol Ther 2020, 108, 1299-1307, doi:10.1002/cpt.1980.
  2. Kim, H.; Kim, D.H.; Kim, D.M.; Kholinne, E.; Lee, E.S.; Alzahrani, W.M.; Kim, J.W.; Jeon, I.H.; Koh, K.H. Do Nonsteroidal Anti-Inflammatory or COX-2 Inhibitor Drugs Increase the Nonunion or Delayed Union Rates After Fracture Surgery?: A Propensity-Score-Matched Study. J Bone Joint Surg Am 2021, 103, 1402-1410, doi:10.2106/JBJS.20.01663.
  3. Morales, D.R.; Conover, M.M.; You, S.C.; Pratt, N.; Kostka, K.; Duarte-Salles, T.; Fernandez-Bertolin, S.; Aragon, M.; DuVall, S.L.; Lynch, K.; et al. Renin-angiotensin system blockers and susceptibility to COVID-19: an international, open science, cohort analysis. Lancet Digit Health 2021, 3, e98-e114, doi:10.1016/S2589-7500(20)30289-2.
  4. Seo, W.W.; Seo, S.I.; Kim, Y.; Yoo, J.J.; Shin, W.G.; Kim, J.; You, S.C.; Park, R.W.; Park, Y.M.; Kim, K.J.; et al. Impact of pitavastatin on new-onset diabetes mellitus compared to atorvastatin and rosuvastatin: a distributed network analysis of 10 real-world databases. Cardiovasc Diabetol 2022, 21, 82, doi:10.1186/s12933-022-01524-6.
  5. Suchard, M.A.; Schuemie, M.J.; Krumholz, H.M.; You, S.C.; Chen, R.; Pratt, N.; Reich, C.G.; Duke, J.; Madigan, D.; Hripcsak, G.; et al. Comprehensive comparative effectiveness and safety of first-line antihypertensive drug classes: a systematic, multinational, large-scale analysis. Lancet 2019, 394, 1816-1826, doi:10.1016/S0140-6736(19)32317-7.
  6. Vashisht, R.; Jung, K.; Schuler, A.; Banda, J.M.; Park, R.W.; Jin, S.; Li, L.; Dudley, J.T.; Johnson, K.W.; Shervey, M.M.; et al. Association of Hemoglobin A1c Levels With Use of Sulfonylureas, Dipeptidyl Peptidase 4 Inhibitors, and Thiazolidinediones in Patients With Type 2 Diabetes Treated With Metformin: Analysis From the Observational Health Data Sciences and Informatics Initiative. JAMA Netw Open 2018, 1, e181755, doi:10.1001/jamanetworkopen.2018.1755.

Reviewer 2 Report

Is this the expansion of the same cohort with results published in 2018? (Ref 4). Jung’s research is similar to this one with four times smaller number of patients.

I have no objection to the methods and statistics used, although it raises some interesting questions. With regard to the number of patients and similar results already published, I would suggest looking into POP risk factors for some of the specific cancer types. I.e. gastric and lung since there were approx. 100 or more patients with pneumonia.

Line 38-40. Refers to commentary by Walder (ref. 1) on the article by Russotto et al. It is by no means an epidemiological study therefore cannot be used as a reference for the percentage of short-term mortality by postoperative pneumonia.

Line 45. Refers to cancer patients having higher incidence of pneumonia than the general population. Ref. 4 (Jung 2018) investigated only cancer patients, similar to this research... no comparison to the general population.

If the above is adequately addressed, I have no objection to the publication of the article.

Author Response

Reviewer 2

  1. Is this the expansion of the same cohort with results published in 2018? (Ref 4). Jung’s research is similar to this one with four times smaller number of patients.

Reply: We thank the reviewer for the appropriate comments. We assure you that our study is completely different from the study you have mentioned in terms of the composition of the patient population. Previous studies have integrated patient data acquired from five hospitals in South Korea, but our study used data from a common data model (CDM) that standardizes medical records, which is very uniform in subject distribution and quality of practice. We further believe that it is very important to use medical big data at the stage of selecting a topic of research. However, we have used previous publications as a reference, especially regarding the research methodology.

  1. I have no objection to the methods and statistics used, although it raises some interesting questions. With regard to the number of patients and similar results already published, I would suggest looking into POP risk factors for some of the specific cancer types. I.e. gastric and lung since there were approx. 100 or more patients with pneumonia.

Reply: We thank the reviewer for the valuable comments. Based on your comment, we have conducted subgroup analysis focusing on lung cancer and gastric cancer and added the results in the results section and in Table S1 as follows:

“Table S1 shows the results of subgroup analysis of POP risk factors for lung and gastric cancers. In univariate Cox regression analysis, POP-related variables associated with lung cancer were old age, male sex, hypertension, chronic pulmonary disease, mood disorder, and cerebrovascular disease. However, in multivariate analysis, only old age, mood disorder, and cerebrovascular disease were associated with POP occurrence. In univariate Cox regression analysis of gastric cancer, the variables associated with POP were old age, male sex, hypertension, renal disease, chronic pulmonary disease, mood disorder, Parkinson's disease, and cerebrovascular disease. In addition, multivariate analysis of gastric cancer showed that older age, male sex, chronic pulmonary disease, and mood disorder were associated with POP occurrence.”

Table S1. Risk factors for postoperative pneumonia in lung cancer and gastric cancer.

Variables

Lung cancer

Gastric cancer

Univariate

Multivariate

Univariate

Multivariate

Hazard ratio
(95% CI)

p-value

Hazard ratio
(95% CI)

P-value

Hazard ratio
(95% CI)

P-value

Hazard ratio
(95% CI)

p-value

Age

1.05
(1.03–1.07)

<0.001

1.05
(1.03–1.07)

<0.001

1.06
(1.04–1.07)

<0.001

1.05
(1.03–1.07)

<0.001

Sex (Ref: Female)

1.66
(1.03–2.69)

0.038

2.30
(1.50–3.50)

<0.001

2.32
(1.51–3.56)

<0.001

Hypertension

2.58
(1.13–5.91)

0.025

2.28
(1.33–3.91)

0.003

Renal disease

2.87
(1.26–6.51)

0.012

Chronic pulmonary disease

2.12
(1.24–3.64)

0.006

3.69
(1.93–7.04)

<0.001

2.39
(1.24–4.61)

0.009

Mood disorder

3.87
(1.22–12.25)

0.021

3.45
(1.09–10.94)

0.036

4.20
(1.34–13.21)

0.014

4.46
(1.37–14.51)

0.013

Parkinson disease

4.09
(1.01–16.53)

0.048

Cerebrovascular disease

3.36
(1.69–6.69)

0.001

2.18
(1.07–4.43)

0.032

2.91
(1.67–5.08)

<0.001

CI: confidence interval

  1. Line 38-40. Refers to commentary by Walder (ref. 1) on the article by Russotto et al. It is by no means an epidemiological study therefore cannot be used as a reference for the percentage of short-term mortality by postoperative pneumonia.

Reply: Thank you for the insightful comment. Based on your advice, we have modified the beginning of the introduction as follows:

“Postoperative pneumonia (POP) is associated with increased morbidity, mortality, and costs [1,2]. This potentially life-threatening complication occurs in up to 9% patients undergoing high-risk surgery [3].”

  1. Line 45. Refers to cancer patients having higher incidence of pneumonia than the general population. Ref. 4 (Jung 2018) investigated only cancer patients, similar to this research... no comparison to the general population.

Reply: Thank you for pointing it out. We apologize for the overlook. Accordingly, we have deleted the erroneous statement as it did not seem to fit the context.

Round 2

Reviewer 1 Report

I have no additional questions, directly for the authors.

Reviewer 2 Report

The authors responded to the comments made during the initial revision.

The paper has been adequately improved.

In that sense, I have no further objections to publishing the article.